# FairVFL: A Fair Vertical Federated Learning Framework with Contrastive Adversarial Learning

**Tao Qi**
Tsinghua University
Beijing, China
taoqi.qt@gmail.com

**Fangzhao Wu**[*]
Microsoft Research Asia
Beijing China
wufangzhao@gmail.com

**Chuhan Wu**
Tsinghua University
Beijing, China
wuchuhan15@gmail.com

**Lingjuan Lyu**
Sony AI
Tokyo, Japan
Lingjuan.Lv@sony.com

**Tong Xu**
USTC
Hefei, China
tongxu@ustc.edu.cn

**Hao Liao**
Shenzhen University
Shenzhen, China
haoliao@szu.edu.cn

**Zhongliang Yang**
Tsinghua University
Beijing, China
yangzl15@tsinghua.org.cn

**Yongfeng Huang**
Tsinghua University
& Zhongguancun Laboratory
yfhuang@tsinghua.edu.cn

**Xing Xie**
Microsoft Research Asia
Beijing China
xingx@microsoft.com

## Abstract

Vertical federated learning (VFL) is a privacy-preserving machine learning paradigm that can learn models from features distributed on different platforms in a privacy-preserving way. Since in real-world applications the data may contain bias on fairness-sensitive features (e.g., gender), VFL models may inherit bias from training data and become unfair for some user groups. However, existing fair machine learning methods usually rely on the centralized storage of fairness-sensitive features to achieve model fairness, which are usually inapplicable in federated scenarios. In this paper, we propose a fair vertical federated learning framework (*FairVFL*), which can improve the fairness of VFL models. The core idea of *FairVFL* is to learn unified and fair representations of samples based on the decentralized feature fields in a privacy-preserving way. Specifically, each platform with fairness-insensitive features first learns local data representations from local features. Then, these local representations are uploaded to a server and aggregated into a unified representation for the target task. In order to learn a fair unified representation, we send it to each platform storing fairness-sensitive features and apply adversarial learning to remove bias from the unified representation inherited from the biased data. Moreover, for protecting user privacy, we further propose a contrastive adversarial learning method to remove private information from the unified representation in server before sending it to the platforms keeping fairness-sensitive features. Experiments on three real-world datasets validate that our method can effectively improve model fairness with user privacy well-protected.

## 1 INTRODUCTION

In recent years, the explosion of data volume has enhanced the performance of machine learning (ML) models on many tasks, e.g., personalized recommendation and search [41, 50, 2]. In many real-world

---

[*]The corresponding author.

36th Conference on Neural Information Processing Systems (NeurIPS 2022).

scenarios, feature fields of the same sample may be decentralized on different platforms [48, 29]. For instance, video watching behaviors and search engine behaviors of the same user usually spread across different platforms [42]. To learn models from various input features, existing machine learning methods usually rely on centralized storage of different feature fields [31, 21]. However, the data on different platforms is usually privacy-sensitive, and cannot be centralized due to privacy concerns and risks [47, 36]. Thus, vertical federated learning (VFL), which can utilize decentralized features for unified model learning in a privacy-preserving way, has become more and more important [53, 9, 7].

Existing VFL methods usually communicate intermediate results instead of raw data across platforms to enable models utilizing decentralized feature fields [53, 9]. For example, Yang et al. [49] proposed a vertical federated framework for logistic regression. They first calculated local results based on local feature fields, and then shared encrypted local results with a server to make predictions and learn parameters. Che et al. [7] proposed a vertical federated multi-view learning framework to integrate decentralized medical data. They locally encoded medical feature fields kept by multiple hospitals into representation vectors and uploaded them to the server for diagnosis prediction. However, since real-world data usually encodes bias on fairness-sensitive features (e.g., genders and ages) [27, 51], VFL models may inherit bias from data and become unfair to some user groups [45, 46].

Due to the importance of fairness in machine learning, fair ML methods have attracted increasing attention in recent years [45, 46, 27, 51, 26]. For example, Zafar et al. [51] proposed a fairness-aware optimization framework to improve the model fairness, which can constrain the difference between prediction distributions for different user groups. Li et al. [26] proposed a fair recommendation framework, which quantifies the recommendation unfairness on different user groups as a penalty for item re-ranking. In general, most of the existing fair ML methods rely on the centralized storage of fairness-sensitive features to achieve model fairness. However, in vertical federated learning, feature fields are decentralized on different platforms [43, 14], making it difficult to apply existing fair ML methods to improve the fairness of VFL models. Thus, how to design a fair VFL framework that can protect user privacy and meanwhile improve model fairness, is a rarely studied problem.

In this paper, we propose a fair vertical federated learning framework (named *FairVFL*), which can improve the fairness of VFL models in a privacy-preserving way. In *FairVFL*, we partition data into two groups based on their fairness sensitivities, i.e., fairness-insensitive features that can be used for the target task and fairness-sensitive features (e.g., genders) that should be causally irrelevant to model predictions. Along this line, the core of *FairVFL* is to learn unified and fair representations of samples based on their features distributed across different platforms with user privacy well-protected. Specifically, platforms with fairness-insensitive features first learn local data representations, which are then uploaded to servers to build unified representations. Since unified representations encode rich information of raw data, they can be used for the target task without access to the raw data. Besides, unified representations may inherit bias towards sensitive features (e.g., gender stereotype) from implicit feature correlations in biased data and hurt model fairness [45, 27, 46]. In order to further learn fair unified representations, we apply adversarial learning to unified representations to remove bias encoded in them. Due to the decentralized storage of feature fields, unified representations are sent to platforms with fairness-sensitive features to learn adversarial gradients. Besides, since unified representations may still encode some private information of raw data [40, 34, 33], directly sharing them among platforms may incur privacy problems. Thus, we further use contrastive adversarial learning to remove private information from the unified representations before sending them to other platforms. Extensive experiments on three real-world datasets show that our *FairVFL* approach can effectively improve the fairness of VFL models and meanwhile the privacy is well protected.

## 2 RELATED WORK

### 2.1 Vertical Federated Learning

Vertical federated learning is a privacy-preserving paradigm for training ML models from feature fields that are kept by different platforms that share the same data ID space, and has been widely studied in recent years [43, 7, 14, 8]. For example, Wu et al. [43] proposed a privacy-preserving ad CTR prediction framework with vertical federated learning. They proposed to coordinate local platforms to learn behavior representations from local features and share the behavior representations with the server to model user interest in ads. However, since real-world data usually encode bias on fairness-sensitive features (e.g., genders) [45, 46], VFL models may inherit bias from data and make

unfair decisions for some user groups (e.g., minority users). To improve model fairness in VFL, Liu et al. [28] formulated the model performance gap for different user groups as a fairness regularization and incorporated it into the federated optimization objective. However, their method needs to share the data grouping information with the server, which may also leak user privacy on fairness-sensitive features (e.g., gender). Different from these methods, we propose a unified framework that can improve the fairness of VFL models which applies the adversarial learning to learn fair models and employ a novel contrastive adversarial learning method to protect user privacy.

## 2.2 Fair Machine Learning

With the increasing impacts of ML techniques on our society, the fairness of machine learning models has attracted substantial attention [45, 22, 26, 46]. Existing fair machine learning methods are usually designed to learn unbiased models with respect to various fairness-sensitive features (e.g., genders). Some of them applied pre-processing techniques [5, 32, 13, 6] (e.g., resampling) or post-processing techniques [26, 3, 22, 18] (e.g., re-ranking) to improve model fairness. However, these methods usually independently optimized model accuracy and fairness, which may only achieve a sub-optimal trade-off between accuracy and fairness [45, 46]. To better improve model fairness, many works have been proposed to jointly optimize model accuracy and fairness during model training [51, 15, 27, 46]. However, most of the existing fair ML methods rely on the centralized storage of feature fields, making them difficult to be applied in vertical federated learning. Different from these methods, we propose a fair vertical federated learning framework, which applies adversarial learning to improve the fairness of VFL models based on decentralized feature fields.

## 3 Methodology

### 3.1 Problem Definition of FairVFL

In VFL, different feature fields of the same samples are decentralized on multiple platforms. From a fairness perspective, these data can be partitioned into two groups, i.e., fairness-insensitive features and fairness-sensitive features. The former can be taken by models as the inputs while the latter are expected to be causally irrelevant to the model predictions. Without loss of generality, we assume that there are $m$ types of fairness-sensitive features, and we denote the platform keeping the $i$-th one with type $d_i^a$ as $P_i^a$.[2] Besides, we assume that fairness-insensitive feature fields are decentralized on $n$ platforms, and we denote the $i$-th platform as $P_i^b$. Moreover, we assume that there is a task platform $P^t$ keeping labels $y$ of samples on the target task. Following existing works [43, 50], we assume that there is a trust-worthy server $P^w$ for information aggregation.

Next, we present a formal definition of privacy protection and fairness. The privacy constraint requires that the private information in the local data of a platform should not be disclosed to another platform. Thus, the raw data of a local platform cannot be disclosed and the intermediate results shared with other platforms should be carefully protected. Besides, in this paper, we define model fairness from a casual view [27]. The counterfactual fairness requires that the model prediction for a sample should always hold if we only change the fairness-sensitive features. According to the theoretical analysis in Li et al. [27], eliminating fairness-sensitive feature information in the data representation that is used for the model prediction can achieve casual fairness. We remark that even though fairness-sensitive features are not taken as the inputs, the model may also inherit bias related to sensitive features by mining inherent shortcuts from data. [45, 27, 46]. The goal of fair vertical federated learning is to achieve a good trade-off among performance, privacy, and fairness, i.e., improving the fairness of VFL models with minimal performance drop in a privacy-preserving way.

### 3.2 Federated Model Serving

The main framework of *FairVFL* to utilize decentralized feature fields is to encode them into a unified representation (Fig. 1). The ML model in *FairVFL* is partitioned into three parts, i.e., multiple local models $\{\mathcal{M}_i^l | i = 1, ..., n\}$, an aggregation model $\mathcal{M}^w$ and a task-specific model $\mathcal{M}^t$, where $\mathcal{M}_i^l$ is the $i$-th local model maintained by the platform $P_i^b$. In *FairVFL*, local models are used to encode

---

[2]To simplify the presentation, we use different symbols to denote platforms storing different fairness-sensitive feature fields. We remark that a platform can store multiple sensitive feature fields in practice.

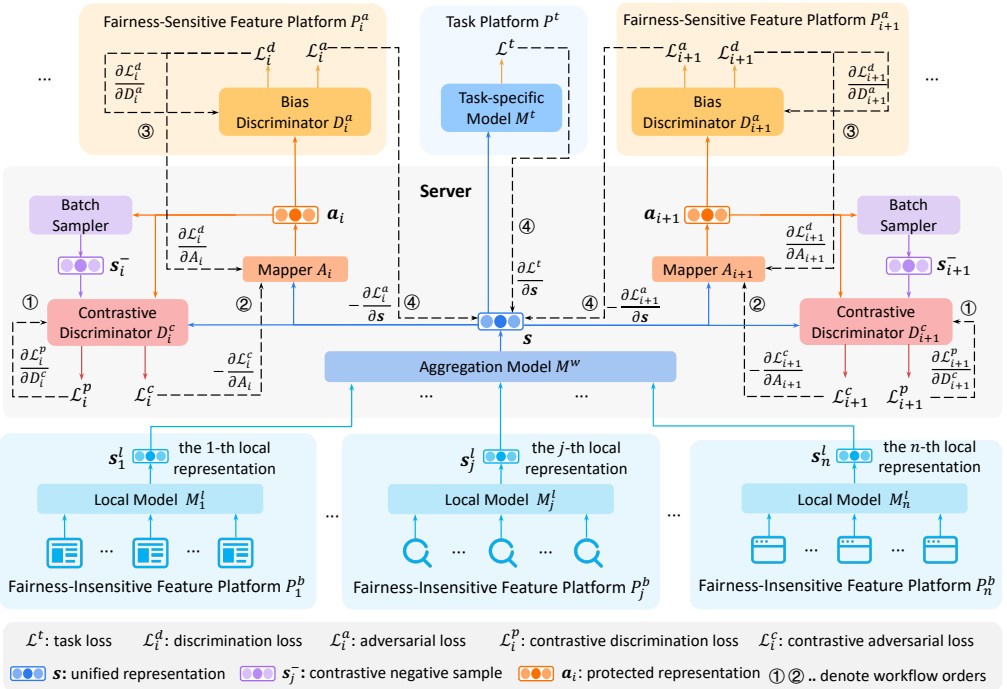

Figure 1: Framework of our fair vertical federated learning method (FairVFL).

local fairness-insensitive features into local representations. We remark that *FairVFL* is a general framework and the architecture of a specific local model can be adjusted based on the input data. For example, the local models for encoding textual search logs can be implemented by transformers [39] or PLMs [10]. When the task platform $P^t$ needs to serve a target sample $s$, $P^t$ will first distribute the ID of the sample to fairness-insensitive platforms $\{P_i^b | i = 1, ..., n\}$. In the $i$-th platform $P_i^b$, the local model $\mathcal{M}_i^l$ encodes fairness-insensitive features of the target sample $s$ stored in $P_i^b$, and builds a local representation $\mathbf{s}_i^l$. Thus, we can obtain multiple local representations $\{\mathbf{s}_i^l | i = 1, ..., n\}$, which encode different fairness-insensitive feature fields of the target sample. Following Wu et al. [43], these local representations are then uploaded to the server $P^w$ for task prediction.

The aggregation model $\mathcal{M}^w$ is maintained by the server $P^w$. It is used to aggregate information in local representations into a unified representation $\mathbf{s}$. Specifically, different feature fields usually have inherent relatedness, mining which can enhance the model for the target task [2, 44]. Thus, we first apply a multi-head self-attention network [39] to capture relatedness among local representations, where the contextual representation of $\mathbf{s}_i^l$ is denoted as $\hat{\mathbf{s}}_i$. Then we apply an attention network to contextual representations to model their relative importance for representing raw data and build the unified representation $\mathbf{s}$.[3] Since $\mathbf{s}$ can encode various information in decentralized fairness-insensitive feature fields, we further upload the unified representation $\mathbf{s}$ instead of raw data to the task platform $P^t$ to provide information for the target task. (We apply LDP to protect $\mathbf{s}$ before sending it to $P^t$.)

The task-specific model $\mathcal{M}^t$ is maintained by the task-specific platform $P^t$. It is used to utilize information in the unified representation $\mathbf{s}$ to make prediction for the target task: $\hat{y} = \mathcal{M}^t(\mathbf{s})$, where $\hat{y}$ is the model prediction. The architecture of the task-specific model $\mathcal{M}^t$ is also adjusted based on the target task (e.g., risk prediction). For instance, for the classification task, we can employ a dense network to implement $\mathcal{M}^t$. In this way, decentralized fairness-insensitive feature fields of a target sample can be utilized by *FairVFL* for the target task without the disclosure of raw data.

### 3.3 Fair Representation Learning in *FairVFL*

As introduced in Sec. 3.2, the task prediction for each sample is computed based on their unified representation $\mathbf{s}$. Thus, the core of *FairVFL* to achieve counterfactual fairness is to learn a fair

---

[3]The aggregation model can be also implemented by other pooling networks, like a dense network.

and unified representation **s** that does not encode fairness-sensitive bias. Although the fairness-sensitive features are not model inputs, the model may still encode bias related to them by mining inherent shortcuts to these features from data. Thus, we apply adversarial learning to prevent unified representations from encoding fairness-sensitive bias. Vanilla adversarial learning techniques usually rely on centralized storage of input data and labels of fairness-sensitive attributes to learn adversarial gradients for model training. However, in vertical federated scenarios, feature fields are decentralized on different platforms, which brings challenges to the application of existing methods. An intuitive solution is sharing the unified representation with platforms that store observed fairness-sensitive features $\{P_i^a | i = 1, 2, ..., m\}$ to learn corresponding adversarial gradients. However, unified representations usually encode some private information, and directly sharing them across different platforms may lead to user privacy leakage. To tackle this challenge, we propose a contrastive adversarial learning technique to protect user privacy by modifying unified representations.

As shown in Fig. 2, the core idea of the contrastive adversarial learning is to learn a protected representation that only encodes information on a certain fairness-sensitive feature. The protected representation can be further shared with the platform keeping the corresponding sensitive feature to learn adversarial gradients without leaking user privacy. Specifically, for each fairness-sensitive feature $d_i^a$, we first map unified representation **s** into a protected representation $\mathbf{a}_i$: $\mathbf{a}_i = A_i(\mathbf{s})$, where $A_i$ is a mapper based on MLP and **s** is the preimage of $\mathbf{a}_i$. $\mathbf{a}_i$ is designed to only retain information of **s** on the fairness-sensitive feature $d_i^a$ and eliminate other user privacy. To achieve this goal, we propose the contrastive adversarial learning method to make the preimage of $\mathbf{a}_i$ indistinguishable among unified representations with the same feature $d_i^a$. Given representations of samples in the same training batch $\{(\mathbf{a}_i^j, \mathbf{s}^j) | j = 1, 2, .., E\}$, we first rank these samples based on the relevance between their protected representations $\mathbf{a}_i^j$ and $\mathbf{a}_i$: $r_j = \mathbf{a}_i \cdot \mathbf{a}_i^j$, where $\mathbf{a}_i^j$ and $\mathbf{s}^j$ are the protected and unified representation of the $j$-th sample, $r_j$ is the relevance score, and $E$ is the batch size.[4] Intuitively, highly relevant data samples are likely to share the same fairness-sensitive feature $d_i^a$. Thus, from the top $E_i$ samples ranked by relevance scores, we can randomly select a unified representation $\mathbf{s}_i^-$ that is highly likely to be the same with **s** in $d_i^a$, where $E_i$ is a hyper-parameter. $\mathbf{s}_i^-$ is used as the negative sample in contrastive learning to learn a contrastive discriminator $D_i^c$ to classify preimage of $\mathbf{a}_i$:

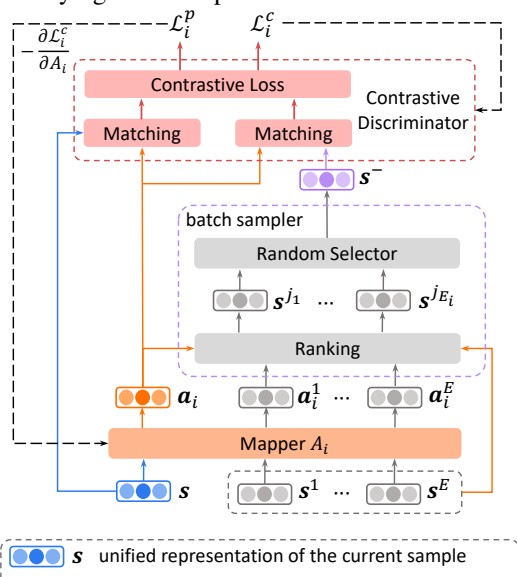

Figure 2: The detailed workflow of contrastive adversarial learning.

$$\mathcal{L}_i^p = -\frac{1}{|\Omega|} \sum_{x \in \Omega} \log \frac{\exp(D_i^c(\mathbf{a}_i, \mathbf{s}))}{\exp(D_i^c(\mathbf{a}_i, \mathbf{s})) + \exp(D_i^c(\mathbf{a}_i, \mathbf{s}_i^-))}, \quad (1)$$

where $D_i^c(\cdot)$ is implemented by an MLP network, $\Omega$ is the training set and $x$ is a data sample in $\Omega$. Optimal discriminator $\hat{D}_i^c$ is learned by minimizing $\mathcal{L}_i^p$ with fixed representations $\mathbf{s}, \mathbf{s}_i^-, \mathbf{a}_i$:

$$\hat{D}_i^c = \arg \min_{D_i^c} \mathcal{L}_i^p. \quad (2)$$

Finally, the optimal discriminator $\hat{D}_i^c$ is used to learn contrastive adversarial loss $\mathcal{L}_i^c$ and the contrastive adversarial gradients $\frac{\partial \mathcal{L}_i^c}{\partial \mathbf{a}_i}$ on $\mathbf{a}_i$ with the fixed optimal discriminator $\hat{D}_i^c$, **s** and $\mathbf{s}_i^-$:

$$\mathcal{L}_i^c = -\frac{1}{|\Omega|} \sum_{x \in \Omega} \log \frac{\exp(\hat{D}_i^c(\mathbf{a}_i, \mathbf{s}))}{\exp(\hat{D}_i^c(\mathbf{a}_i, \mathbf{s})) + \exp(\hat{D}_i^c(\mathbf{a}_i, \mathbf{s}_i^-))}. \quad (3)$$

---

[4]The relevance score for the current data sample is masked as $-\infty$.

The contrastive adversarial gradient $\frac{\partial \mathcal{L}_i^c}{\partial \mathbf{a}_i}$ on $\mathbf{a}_i$ is further back-propagated to the mapper $A_i$:

$$\frac{\partial \mathcal{L}_i^c}{\partial A_i} = \frac{\partial \mathcal{L}_i^c}{\partial \mathbf{a}_i} \frac{\partial \mathbf{a}_i}{\partial A_i}, \quad \mathbf{a}_i = A_i(\mathbf{s}), \tag{4}$$

where negative contrastive adversarial gradient $-\frac{\partial \mathcal{L}_i^c}{\partial A_i}$ on $A_i$ is used for model updating, and representations $\mathbf{s}$ and $\mathbf{s}_i^-$ are fixed and not tuned by the contrastive adversarial gradient. In this way, contrastive adversarial gradients can enforce the mapper $A_i$ to protect user private information in $\mathbf{a}_i$.

Since user privacy can be reduced from $\mathbf{a}_i$, we further share it with the platform $P_i^a$ keeping label $\mathbf{y}_i^a$ of the fairness-sensitive feature $d_i^a$ to learn corresponding adversarial gradients. The platform $P_i^a$ first employs a bias discriminator $D_i^a$ to predict the label $\hat{\mathbf{y}}_i^a$ of fairness-sensitive feature $d_i^a$ from the protected representation $\mathbf{a}_i$ and obtain a bias discrimination loss $\mathcal{L}_i^d$ as follows:

$$\mathcal{L}_i^d = -\frac{1}{|\Omega|} \sum_{x \in \Omega} \mathbf{y}_i^a \cdot \log \hat{\mathbf{y}}_i^a, \quad \hat{\mathbf{y}}_i^a = D_i^a(\mathbf{a}_i), \tag{5}$$

where $D_i^a$ is implemented by an MLP network. Moreover, besides private information, contrastive adversarial learning may remove bias information from $\mathbf{a}_i$ and hurt fair representation learning. Thus, we minimize bias discrimination loss $\mathcal{L}_i^d$ to optimize both bias discriminator $D_i^a$ and mapper $A_i$:

$$\hat{D}_i^a, \hat{A}_i = \arg \min_{D_i^a, A_i} \mathcal{L}_i^d = \arg \min_{D_i^a, A_i} -\mathbf{y}_i^a \cdot \log D_i^a(A_i(\mathbf{s})), \tag{6}$$

where $\hat{D}_i^a$ is the optimal bias discriminator, $\hat{A}_i$ is the $i$-th optimal mapper, and the unified representation $\mathbf{s}$ is fixed. Finally, for the $i$-th fairness-sensitive feature $d_i^a$, we can obtain the adversarial loss $\mathcal{L}_i^a$ and corresponding adversarial gradients $\frac{\partial \mathcal{L}_i^a}{\partial \mathbf{s}}$ on unified representation $\mathbf{s}$:

$$\mathcal{L}_i^a = -\frac{1}{|\Omega|} \sum_{x \in \Omega} \mathbf{y}_i^a \cdot \log \hat{\mathbf{y}}_i^a, \quad \hat{\mathbf{y}}_i^a = \hat{D}_i^a(\hat{A}_i(\mathbf{s})), \tag{7}$$

where the optimal bias discriminator and mapper are fixed when optimizing the unified representation $\mathbf{s}$. The negative adversarial gradient $-\frac{\partial \mathcal{L}_i^a}{\partial \mathbf{s}}$ is further used to reduce fairness-sensitive bias encoded in the unified representation $\mathbf{s}$ to achieve counterfactual fairness. Besides, we can obtain the task loss $\mathcal{L}^t$ based on task label $y$ and predicted label $\hat{y}$. The task loss function is also adjusted based on the target task. Finally, we can formulate the overall loss $\mathcal{L}$ as:

$$\mathcal{L} = \mathcal{L}^t - \sum_{i=1}^m \lambda_i \mathcal{L}_i^a - \sum_{i=1}^m \gamma_i \mathcal{L}_i^c, \tag{8}$$

where $\mathcal{L}^t$, $\mathcal{L}^a$ and $\mathcal{L}^c$ represents the training objective for model performance, fairness and privacy respectively, $\lambda_i$ is the weight of loss $\mathcal{L}_i^a$ and $\gamma_i$ is the weight of loss $\mathcal{L}_i^c$.

## 3.4 Federated Model Training

Next, we will introduce how we train model on decentralized fairness-sensitive and insensitive feature fields. First, for the task loss $\mathcal{L}^t$, the task platform $P^t$ can directly calculate gradient $\frac{\partial \mathcal{L}^t}{\partial \mathcal{M}^t}$ on task model and gradient $\frac{\partial \mathcal{L}^t}{\partial \mathbf{s}}$ on unified representation $\mathbf{s}$. $\frac{\partial \mathcal{L}^t}{\partial \mathcal{M}^t}$ is used to update $\mathcal{M}^t$ and $\frac{\partial \mathcal{L}^t}{\partial \mathbf{s}}$ is distributed to the server $P^w$ for the further model updating. Second, it is usually difficult to obtain the optimal discriminator in adversarial learning. Following Goodfellow et al. [16], we optimize the discriminator for a single step based on discrimination gradients before executing adversarial learning. Specifically, for the contrastive adversarial learning, the sever first learns contrastive discrimination gradient $\frac{\partial \mathcal{L}_i^p}{\partial D_i^c}$ on contrastive discriminator $D_i^c$ to update it. Then the server utilizes updated $D_i^c$ to learn contrastive adversarial gradient $\frac{\partial \mathcal{L}_i^c}{\partial A_i}$ on mapper $A_i$ to update it. Third, for adversarial learning on each fairness-sensitive feature $d_i^a$, platform $P_i^a$ first calculates bias discrimination gradient $\frac{\partial \mathcal{L}_i^d}{\partial D_i^a}$ on the bias discriminator $D_i^a$ and $\frac{\partial \mathcal{L}_i^d}{\partial \mathbf{a}_i}$ on protected representation $\mathbf{a}_i$. The former is used to optimize $D_i^a$. The latter is distributed to the server $P^w$ to calculate gradient on $A_i$: $\frac{\partial \mathcal{L}_i^a}{\partial A_i} = \frac{\partial \mathcal{L}_i^a}{\partial \mathbf{a}_i} \frac{\partial \mathbf{a}_i}{\partial A_i}$, which

is used to optimize $A_i$. Next, the platform $P_i^a$ calculates the adversarial gradient $\frac{\partial \mathcal{L}_i^a}{\partial \mathbf{a}_i}$ on $\mathbf{a}_i$. $\frac{\partial \mathcal{L}_i^a}{\partial \mathbf{a}_i}$ is further distributed to $P^w$ to calculate adversarial gradients $\frac{\partial \mathcal{L}_i^a}{\partial \mathbf{s}}$ on the unified representation $\mathbf{s}$ based on gradient back propagation: $\frac{\partial \mathcal{L}_i^a}{\partial \mathbf{s}} = \frac{\partial \mathcal{L}_i^a}{\partial \mathbf{a}_i} \frac{\partial \mathbf{a}_i}{\partial \mathbf{s}}$. Until now, the server $P^w$ can further calculate overall gradients $\frac{\partial \mathcal{L}}{\partial \mathbf{s}} = \frac{\partial \mathcal{L}^t}{\partial \mathbf{s}} - \sum_{i=1}^{m} \lambda_i \frac{\partial \mathcal{L}_i^a}{\partial \mathbf{s}}$ on the unified representation $\mathbf{s}$. The server further calculates gradient $\frac{\partial \mathcal{L}}{\partial \mathcal{M}^w}$ on the aggregation model: $\frac{\partial \mathcal{L}}{\partial \mathcal{M}^w} = \frac{\partial \mathcal{L}}{\partial \mathbf{s}} \frac{\partial \mathbf{s}}{\partial \mathcal{M}^a}$ and the gradient $\frac{\partial \mathcal{L}}{\partial \mathbf{s}_i^l}$ on each local representation $\mathbf{s}_i^l$: $\frac{\partial \mathcal{L}}{\partial \mathbf{s}_i^l} = \frac{\partial \mathcal{L}}{\partial \mathbf{s}} \frac{\partial \mathbf{s}}{\partial \mathbf{s}_i^l}$. Finally, gradients on local representations are distributed to corresponding platforms $\{P_i^b | i = 1, 2, ..., n\}$ to optimize local models $\{\mathcal{M}_i^l | i = 1, 2, ..., n\}$. The detailed process of a training round in *FairVFL* is in Algorithm 1 (Supplementary).

### 3.5 Discussion on Privacy and Efficiency

**Privacy**: In *FairVFL*, each data platform locally keeps its private data and never shares it with the outside, which can protect user privacy to some extent. Besides, for the vertical federated model learning and inference, the intermediate model results, i.e., data representations and model gradients, need to be exchanged across platforms, which may lead to potential privacy leakage. To tackle this challenge, we propose a contrastive adversarial learning method to protect private information in data representations, which can compare data representations with the same attribute and remove the attribute-irrelated information in them. To further protect private information in gradients, we can apply differential privacy techniques to perturb exchanged gradients as standard VFL methods [47].

**Efficiency:** *FairVFL* needs multiple times of communications in a single training round, which may arouse concerns on communication efficiency. Fortunately, the extra communication costs are usually small. This is because in the extra communication rounds *FairVFL* only needs to exchange protected representations and their gradients, and the extra communication cost is $\mathcal{O}(4mEH)$, where $m$ is the number of fairness-sensitive platforms, $H$ denotes the dimensionality of protected representations, and $E$ denotes the batch size. Since $M$, $E$, and $B$ is usually small in practice, the corresponding cost is usually minor. For the computation efficiency, compared with the standard *VFL* framework, *FairVFL* only increases the computation cost of the server and platforms with fairness-sensitive attributes. They need to extra compute the gradients on the attribute discriminators and contrastive discriminators. Thus, the computation complexities of *FairVFL* and standard *VFL* are $\mathcal{O}(d^3 + D^3)$ and $\mathcal{O}(D^3)$ respectively, where $d$ and $D$ denote the parameter sizes of the discriminators and the main task model. In widely used settings, the discriminator is usually implemented by a small FFNN network, whose parameters are usually much less than the main task model. Thus, the computation efficiency of *FairVFL* is still comparable with standard *VFL* methods.

## 4 Experiment

### 4.1 Datasets and Experimental Settings

We evaluate model performance and fairness on three real-world datasets. The first one is *ADULT* [24], which is a widely used public dataset for fair ML [52, 20, 35, 25]. It contains rich feature fields (e.g., education and social relation) for income prediction. 20,000 randomly selected data samples are used to construct training and validation dataset, and 10,000 randomly selected data samples are used to construct test dataset. Following existing works [45, 46], we treat gender and age as fairness-sensitive features and utilize other feature fields for income prediction. Besides, we randomly partition 12 types of feature fields in *ADULT* into three platforms to simulate the VFL scenarios[5]. The second one is *NEWS*, which is constructed by user logs (i.e., news clicks, search, and web-page browsing logs) on Microsoft News and Bing Search platforms during 6 weeks (June 23th - July 20th, 2019) for news recommendation. These three types of data are decentralized on three platforms under VFL settings. We randomly select 100,000 news impressions in the first three weeks to construct the training and validation set and select 100,000 news impressions in the last week to construct the test set. Besides, we also treat gender and age as fairness-sensitive attributes in *NEWS*. The third one is *CelebA* [30], which is a public face attributes dataset. To simulate the VFL setting, we use the raw images and a part of the raw attributes (i.e., the "Straight Hair" attribute and "Wavy Hair") as the input features. The raw images are stored in a data platform and other input attributes are stored in another platform.

---

[5]*FairVFL* is feature partition agnostic and does not rely on specific partition strategy.

Table 1: Model performance and fairness on *ADULT* and *NEWS*.

| Training Strategy | Model | ADULT | | | | Model | NEWS | | | |
| | | Income Prediction | | Model Fairness | | | News Recommendation | | Model Fairness | |
| | | Accuracy | F1 | Gender F1 | Age F1 | | AUC | nDCG@10 | Gender F1 | Age F1 |
|---|---|---|---|---|---|---|---|---|---|---|
| CenTrain | MLP | 82.15±0.86 | 78.42±0.66 | 78.27±1.60 | 47.47±0.90 | NAML | 64.04±0.13 | 30.80±0.13 | 70.05±0.21 | 20.01±3.08 |
| | TabNet | 82.23±1.02 | 78.50±0.80 | 79.07±1.79 | 49.12±1.24 | LSTUR | 64.68±0.33 | 30.97±0.20 | 70.45±0.37 | 20.00±0.39 |
| | AutoInt | 82.31±1.92 | 78.49±1.50 | 79.22±0.84 | 48.99±1.25 | NRMS | 64.24±0.18 | 30.78±0.11 | 70.25±0.24 | 21.07±0.81 |
| FairGo | MLP | 77.97±1.34 | 78.49±1.25 | 50.92±6.25 | 15.92±3.66 | NAML | 60.73±0.25 | 28.30±0.16 | 54.08±5.97 | 15.93±1.65 |
| | TabNet | 74.40±1.71 | 75.33±1.47 | 50.28±5.32 | 15.63±3.14 | LSTUR | 61.03±0.24 | 28.31±0.15 | 53.57±3.88 | 15.74±1.22 |
| | AutoInt | 76.89±1.50 | 77.31±1.27 | 50.59±4.41 | 15.73±2.27 | NRMS | 61.49±0.37 | 28.83±0.31 | 53.81±2.94 | 16.45±1.49 |
| FairSM | MLP | 77.57±1.39 | 78.31±1.06 | 50.66±6.43 | 15.55±3.10 | NAML | 60.59±0.19 | 28.15±0.16 | 54.13±5.38 | 15.74±1.52 |
| | TabNet | 74.04±1.66 | 75.07±1.39 | 50.61±4.44 | 15.98±3.14 | LSTUR | 61.11±0.54 | 28.33±0.34 | 53.16±4.75 | 15.24±1.84 |
| | AutoInt | 76.30±2.56 | 76.88±2.05 | 50.53±5.02 | 15.50±3.10 | NRMS | 61.78±0.31 | 28.95±0.22 | 54.10±2.42 | 15.91±1.99 |
| FairRec | MLP | 77.59±1.42 | 78.08±1.24 | 50.94±7.15 | 15.75±4.62 | NAML | 60.69±0.22 | 28.21±0.17 | 54.47±2.79 | 15.67±1.90 |
| | TabNet | 74.89±1.76 | 75.65±1.61 | 50.35±5.79 | 15.61±3.18 | LSTUR | 60.99±0.78 | 28.31±0.47 | 53.36±1.37 | 15.65±0.93 |
| | AutoInt | 76.60±1.91 | 77.17±1.54 | 50.43±7.01 | 15.83±3.96 | NRMS | 61.44±0.16 | 28.76±0.21 | 53.60±2.84 | 16.26±2.39 |
| VFL | MLP | 81.47±2.14 | 77.82±1.57 | 79.05±0.91 | 47.48±1.25 | NAML | 63.93±0.45 | 30.75±0.45 | 69.72±0.48 | 20.09±0.86 |
| | TabNet | 81.77±1.72 | 78.09±1.35 | 78.75±0.77 | 48.36±1.32 | LSTUR | 64.39±0.32 | 30.85±0.19 | 70.07±0.37 | 19.92±1.63 |
| | AutoInt | 81.65±1.52 | 78.02±1.17 | 79.07±1.42 | 47.98±1.51 | NRMS | 64.38±0.13 | 30.93±0.11 | 70.67±0.23 | 21.41±0.66 |
| FairVFL | MLP | 76.74±2.64 | 77.87±2.14 | 50.31±4.99 | 15.50±4.33 | NAML | 60.41±0.18 | 27.95±0.18 | 53.38±4.40 | 15.55±1.41 |
| | TabNet | 75.51±0.69 | 76.06±0.60 | 50.72±5.72 | 15.48±2.46 | LSTUR | 60.98±0.28 | 28.25±0.36 | 53.51±3.41 | 15.23±0.94 |
| | AutoInt | 76.19±0.99 | 76.86±0.85 | 50.53±4.48 | 15.22±2.93 | NRMS | 61.43±0.13 | 28.81±0.08 | 53.33±2.35 | 15.98±1.94 |

The target task is to classify whether the input sample has the attribute "Smiling". We also treat gender (the "Male" attribute) as the target fairness-sensitive attribute.

In *FairVFL*, the local representations and unified representations are 400-dimensional. The protected representations for gender and age are 32- and 64-dimensional, respectively. The weights of contrastive adversarial loss for different sensitive features are set to 0.25. The weights of adversarial loss for gender and age are set to $\frac{2}{3}$ and $\frac{2}{3}$ on *NEWS*, respectively. Besides, the weights of adversarial loss for gender and age are set to $1e2$ and $1e1$ on *ADULT*, respectively. We exploit Adam algorithm [23] for model optimization with 1e-4 learning rate. The size of the mini-batch for model training is set to 32. Besides, for both age and gender, we set $E_i$ to 5 for simplification. We also use the dropout technique [38] with a 0.2 drop probability to alleviate model overfitting. Hyper-parameters are selected based on the validation dataset. Codes are available in https://github.com/taoqi98/FairVFL.

## 4.2 Performance and Fairness Evaluation

Remark that we can achieve counterfactual fairness by reducing fairness-sensitive features encoded in data representations that are used for the model predictions [27]. Thus, following existing works [12, 4, 46, 27], we utilize the ensemble of 5 attackers to predict fairness-sensitive features from unified representations **s** to evaluate the independence of fairness-sensitive features and data representations to verify model fairness. If attackers can distinguish fairness-sensitive bias from unified representations then we can say the model is unfair. Thus, a lower F1 score on the fairness-sensitive feature classification task means better model fairness. Besides, we use accuracy and F1 score to evaluate the classification task on *ADULT* and *CelebA*. Moreover, following these works [1, 41], we use AUC and nDCG@10 to evaluate the news recommendation task on *NEWS*. Each experiment is repeated five times and we report the average performance.

We compare our *FairVFL* method with several recent representative baseline methods for fair representation learning: (1) *FairGo* [46]: a user-centric fair representation learning framework that utilizes adversarial learning to reduce fairness-sensitive bias information in representations; (2) *FairSM* [27]: a unified framework to achieve counterfactual fairness on representations; (3) *FairRec* [45]: an adversarial representation decomposition framework that can decompose bias and bias-insensitive information. These methods rely on

Table 2: Model performance and fairness on *CelebA*.

| Method | Main Task | | Fairness |
| | Accuracy | F1 | GenderF1 |
|---|---|---|---|
| CenTrain | 0.9180±0.0013 | 0.9180±0.0014 | 0.8706±0.0069 |
| FairGo | 0.9096±0.0086 | 0.9095±0.0088 | 0.5159±0.0570 |
| FairSM | 0.9032±0.0230 | 0.9027±0.0242 | 0.5259±0.0807 |
| FairRec | 0.9070±0.0094 | 0.9067±0.0097 | 0.5241±0.0743 |
| VFL | 0.9192±0.0020 | 0.9174±0.0021 | 0.8727±0.0056 |
| FairVFL | 0.9045±0.0130 | 0.9042±0.0135 | 0.5143±0.0778 |

centralized storage of features to learn fair ML models. Besides, both *FairVFL* and these three methods are general frameworks and can be applied to basic models to improve their fairness. In addition, baseline methods also include a standard vertical federated representation learning method (*VFL*) without fairness criterion [7]. Thus, to compare their effectiveness, we combine them with several basic ML models on the three datasets. Basic models for *ADULT* are implemented by three SOTA

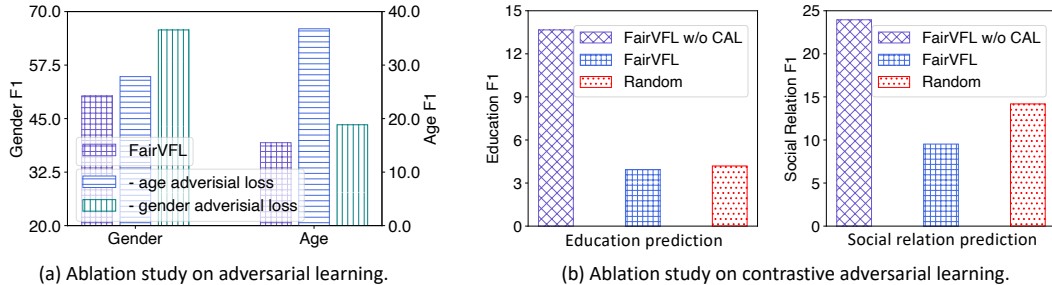

(a) Ablation study on adversarial learning.

(b) Ablation study on contrastive adversarial learning.

Figure 3: Ablation study on adversarial learning and contrastive adversarial learning. CAL denotes contrastive adversarial learning and Random denotes predicting private data randomly.

models for modeling structural features, including *MLP* [17], *TabNet* [11] and *AutoInt* [37]. Basic models for *NEWS* are implemented by SOTA news recommendation models, including *NAML* [41], *LSTUR* [1] and *NRMS* [44]. Moreover, the VFL model for *CelebA* is based on ResNet-10 [19]. Due to space limitations, detailed settings of these basic models are summarized in the Supplementary.

Results on *ADULT* and *NEWS* are summarized in Table 1, and results on *CelebA* are summarized in Table 2, from which we have several observations. First, basic models based on vertical federated training (e.g., MLP+VFL) can achieve similar performance with the same basic models based on centralized training (e.g., MLP). These results show that the vertical federated learning technique can effectively enable ML models to utilize decentralized feature fields for target tasks without the disclose of raw data. Second, models without the protection of fair ML methods (e.g., MLP and MLP+VFL) are biased by fairness-sensitive features. This is because data may encode bias on sensitive features like genders. Models may inherit bias from data and become unfair to some user groups. Third, after applying our *FairVFL* method to basic models, the fairness of these models effectively improves. This is because our *FairVFL* method applies the adversarial learning on bias discrimination to unified representations, which can effectively reduce bias encoded in them and improve model fairness. Fourth, compared with models protected by centralized fair machine learning methods (i.e., *FairGo*, *FairSM* and *FairRec*), models protected by *FairVFL* can achieve similar performance and fairness. These results verify that our *FairVFL* method can effectively improve the fairness of VFL models.

### 4.3 Ablation Study

In this section, we first evaluate the effectiveness of adversarial learning in improving model fairness. Due to space limitations, we only show results on *ADULT* in the following sections. Results are shown in Fig. 3, from which we have several observations. First, after removing adversarial learning on gender discrimination, model fairness on user gender seriously drops. This is because users of the same gender usually have similar behavior patterns and there may exist gender bias in data. Models may inherit gender bias from implicit feature correlations in data and become unfair to some user groups. To tackle this challenge, *FairVFL* applies adversarial learning on bias discrimination to unified representations to reduce gender bias encoded in them. Second, removing adversarial learning for age discrimination also seriously hurts model fairness on user age, which further verifies the effectiveness of adversarial learning for improving model fairness.

Next, we verify the effectiveness of contrastive adversarial learning on privacy protection. Since unified representations include private information of fairness-insensitive features on local platforms, communicating them across platforms may arouse privacy concerns. Motivated by membership inference attack methods, we evaluate the privacy protection by inferring the input features of a unified representation from corresponding protected representations $\{\mathbf{a}_i | i = 1, ..., m\}$ that are shared across platforms $\{P_i^a | i = 1, ..., m\}$. Specifically, based on the *ADULT* dataset, we infer the value of input features from each $\mathbf{a}_i$ via 5 attackers and average the results. A lower inference accuracy (the F1 value) means better privacy protection. We remark that for fairness evaluation, we distinguish bias on fairness-sensitive features from unified representations instead of the shared representations, and these sensitive features are also not the input of representation learning. We only show results on two representative feature fields (i.e., education, and social relation) in Fig. 3 due to space limitations. First, without the protection of contrastive adversarial learning, shared representations are very informative

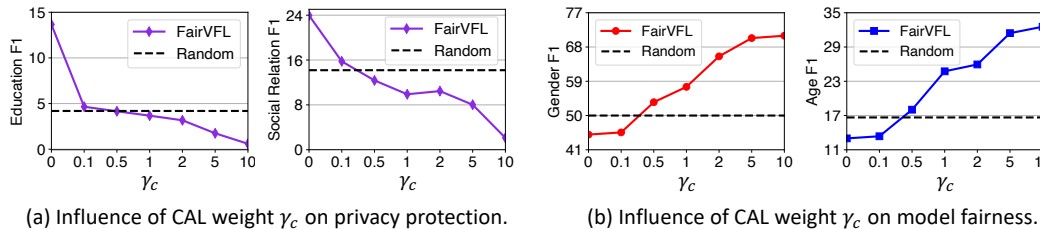

(a) Influence of CAL weight $\gamma_c$ on privacy protection.    (b) Influence of CAL weight $\gamma_c$ on model fairness.

Figure 4: Influence of contrastive adversarial learning weights on privacy protection and model fairness, where contrastive adversarial loss weights for different sensitive features are set to $\gamma_c$.

for inferring private user information. These results verify that directly sharing unified representations across platforms may leak user privacy. Second, contrastive adversarial learning can effectively protect user privacy in protected representations. This is because contrastive adversarial learning enforces protected representations to be indistinguishable for predicting its preimage from unified representations of the same fairness-sensitive features, which can effectively reduce bias-irrelevant personal information in shared representations.

### 4.4 Hyper-Parameter Analysis

Next, we evaluate the influence of the weights of contrastive adversarial learning $\{\gamma_i|i = 1, 2, ..., m\}$ on both privacy protection and model fairness. To simplify the analysis, we set them to the same value (denoted as $\gamma_c$). The influence of $\gamma_c$ on privacy protection and model fairness is shown in Fig. 4, from which we have two observations. First, with the increase of $\gamma_c$, *FairVFL* can achieve better privacy protection ability. This is intuitive since larger $\gamma_c$ makes contrastive adversarial gradients more important to update the mappers, making them can reduce user privacy in unified representations more effectively. Second, larger $\gamma_c$ also hurts model fairness more seriously. This is because larger $\gamma_c$ makes mappers pay more attention to reducing user private information, which may hurt their abilities on retaining information of fairness-sensitive bias information. Thus, the protected representations may contain less bias information and hurt the effectiveness of adversarial learning in improving model fairness. Fortunately, when $\gamma_c$ is small (e.g., 0.25), *FairVFL* is effective in privacy protection since the performance of user privacy inference is close to random. Besides, *FairVFL* can achieve satisfied model fairness (about 0.5 and 0.15 F1 value for gender and age prediction) when $\gamma_c$ is small (e.g., 0.25). This indicates that *FairVFL* can achieve effective privacy protection ability without hurting model fairness and we set $\gamma_c$ to 0.25 in *FairVFL*.

## 5 CONCLUSION

In this paper, we propose a fair vertical federated learning framework (named *FairVFL*), which can improve the fairness of VFL models in a privacy-preserving way. The core of *FairVFL* is to learn fair and unified representations to encode samples based on their decentralized feature fields. Each platform storing fairness-insensitive features first learns local representations from their local features. Then these local representations are uploaded to a server to form a unified representation used for the target task. In order to further learn fair and unified representations, we send them to platforms storing fairness-sensitive features and apply adversarial learning to reduce bias encoded in them. Besides, to protect shared representations from leaking user privacy, we propose a contrastive adversarial learning method to reduce user private information in them before sharing them with other platforms. Experimental results on three datasets show that *FairVFL* is comparable with centralized baseline fair ML methods in terms of both performance and fairness, meanwhile effectively protects user privacy.

### Acknowledgments

This work was supported by the National Key R&D Program of China 2021ZD0113902, the National Natural Science Foundation of China under Grant numbers 82090053, U1936126, and Tsinghua-Toyota Joint Research Fund 20213930033.

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
