# Supplementary

## Federated Fair Model Training

The detailed workflow of our *FairVFL* method is summarized in Algorithm 1.

---

**Algorithm 1** Federated Model Training in *FairVFL*

---

1: **for all** $i$ in $1, 2, ..., n$ **do**
2:  Query $\mathbf{s}_i^l$ from the platform $P_i^b$
3: **end for**
4: Apply $\mathcal{M}^w$ to learn $\mathbf{s}$ from $\{\mathbf{s}_i^l | i = 1, 2, ..., n\}$ on $P^w$
5: Upload $\mathbf{s}$ to $P^t$ to calculate $\frac{\partial \mathcal{L}^t}{\partial \mathbf{s}}$ and $\frac{\partial \mathcal{L}^t}{\partial \mathcal{M}^t}$
6: Use $\frac{\partial \mathcal{L}^t}{\partial \mathcal{M}^t}$ to update $\mathcal{M}^t$ and distribute $\frac{\partial \mathcal{L}^t}{\partial \mathbf{s}}$ to $P^w$
7: **for all** $i$ in $1, 2, ..., m$ **do**
8:  Map $\mathbf{s}$ to $\mathbf{a}_i$ via $A_i$ on $P^w$
9:  Select a contrastive negative sample $\mathbf{s}_i^-$ on $P^w$
10:  Calculate $\frac{\partial \mathcal{L}_i^p}{\partial D_i^c}$ based on Eq. 1 to update $D_i^c$ on $P^w$
11:  Calculate $\frac{\partial \mathcal{L}_i^c}{\partial A_i}$ based on Eq. 3 to update $A_i$ on $P^w$
12:  Recalculate $\mathbf{a}_i$ via updated $A_i$
13:  Upload $\mathbf{a}_i$ from $P^w$ to $P_i^a$
14:  Learn $\frac{\partial \mathcal{L}_i^d}{\partial D_i^a}$ based on Eq. 5 to update $D_i^a$ on $P_i^a$
15:  Learn $\frac{\partial \mathcal{L}_i^d}{\partial \mathbf{a}_i}$ based on Eq. 5 on $P_i^a$ and distribute it to $P^w$
16:  Learn $\frac{\partial \mathcal{L}_i^d}{\partial A_i}$ via $\frac{\partial \mathcal{L}^d}{\partial \mathbf{a}_i}$ on $P^w$ to update $A_i$
17:  Recalculate $\mathbf{a}_i$ via updated $A_i$ and upload it to $P_i^a$
18:  Learn $\frac{\partial \mathcal{L}_i^a}{\partial \mathbf{a}_i}$ based on Eq. 7 on $P_i^a$ and distribute it to $P^w$
19:  Calculate $\frac{\partial \mathcal{L}_i^a}{\partial \mathbf{s}}$ via $\frac{\partial \mathcal{L}_i^a}{\partial \mathbf{a}_i}$ on $P^w$
20: **end for**
21: Calculate $\frac{\partial \mathcal{L}}{\partial \mathbf{s}}$ for $\mathbf{s}$ based on Eq. 8 on $P^w$
22: Calculate $\frac{\partial \mathcal{L}}{\partial \mathcal{M}^w}$ via $\frac{\partial \mathcal{L}}{\partial \mathbf{s}}$ to update $\mathcal{M}^w$ on $P^w$
23: **for all** $i$ in $1, 2, ..., n$ **do**
24:  Calculate $\frac{\partial \mathcal{L}}{\partial \mathbf{s}_i^l}$ via $\frac{\partial \mathcal{L}}{\partial \mathbf{s}}$ on $P^w$ and distribute it to $P_i^b$
25:  Calculate $\frac{\partial \mathcal{L}}{\partial \mathcal{M}_i^l}$ via $\frac{\partial \mathcal{L}}{\partial \mathbf{s}_i^l}$ to update $\mathcal{M}_i^l$ on $P_i^b$
26: **end for**

---

## Experimental Settings

Next, we will introduce the basic models used for model training in details. Since *ADULT* is a tabular feature dataset, we implement three basic models for modeling structural features [17] to predict income on *ADULT*: (1) *MLP* [17]: converting user features into embedding vectors and using an MLP network for income prediction. (2) *TabNet* [11]: using attentive feature transformer networks to model relatedness of different features and build local representations. (3) *AutoInt* [37]: applying multi-head self-attention network to feature embeddings to model their interactions and learn their representations. In these three methods, dimensions of feature embeddings are set to 32, and dimensions of local representations are set to 400. In addition, we choose three mainstream news recommendation models as basic models for the news recommendation task on *NEWS*: (1) *NAML* [41]: applying an attentive CNN network to learn behavior representations and an attention network to learn user representations; (2) *LSTUR* [1]: proposing to learn short-term user representations from recent user behaviors and long-term user representations via user ID embeddings; (3) *NRMS* [44]: employing multi-head self-attention networks to learn behavior and user representations. In these three methods, three types of behavior representations (e.g., clicked news) are set to 400-dimensional.

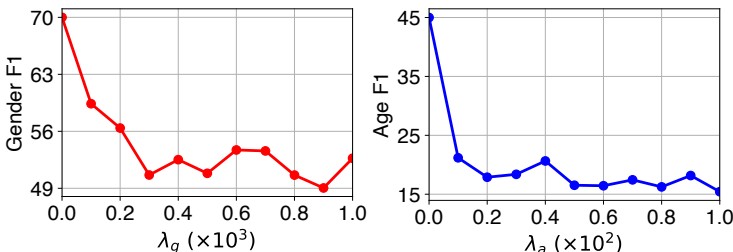

Figure 5: Influence of adversarial learning on model fairness, where $\lambda_g$ and $\lambda_a$ are adversarial learning weights for gender and age discrimination, respectively.

## Influence of Adversarial Learning

Next, we will analyze the influence of the weights of adversarial learning on model fairness. We summarize results in Fig. 5 and have several observations. First, with the increase of $\lambda_g$, model fairness on gender can increase. This is because larger $\lambda_g$ makes models pay more attention to reducing gender bias encoded in unified representations during model training. Second, when $\lambda_g$ is large enough, model fairness on gender becomes stable. This is because when $\lambda_g$ is large enough, gender information in unified representations can be effectively reduced. Third, with the increasing of $\lambda_a$, model fairness on age also first increases and then converges. Similarly, this is because users with similar ages usually have similar behaviors and may encode age bias in real-world data. Larger $\lambda_a$ can more effectively prevent unified representations from encoding age bias from data until age bias is effectively removed.

## Limitations and Future Work

Although *FairVFL* is effective in both model fairness and privacy protection, *FairVFL* may have more communication latency than other baseline methods during model training. This is because the adversarial learning and the contrastive adversarial learning in *FairVFL* make it need to communicate with a fairness-sensitive feature platform multiple times. In our future work, we plan to study an asynchronous communication mechanism to reduce the communication latency of *FairVFL*.