# OpenReview forum: "FairVFL: A Fair Vertical Federated Learning Framework with Contrastive Adversarial Learning"
_NeurIPS.cc/2022/Conference — NeurIPS 2022 Accept_

### Official Review · Reviewer_rSwq · 2022-06-17

**Rating:** 6
**Confidence:** 3
**Soundness:** 2 fair
**Presentation:** 3 good
**Contribution:** 3 good

**Summary:**

This paper proposes a method to learn fair representations in the VFL setting. By fairness, the authors mean that the representations generated by FairVFL should not reflect sensitive attributes, such as age/gender. The authors achieve this goal via a contrastive adversarial learning on the sensitive attributes. The idea of this adversarial training is that, the discriminator aims to learn sensitive attributes, while the feature learner aims to fool the discriminator. Moreover, in order to conceal private information, the adversarial learning also aims to conceal all other information except for the sensitive attribute. Experiments on two real-world datasets show the effectiveness of FairVFL.

**Questions:**

Q1-3: See Weaknesses 1 2 3.

Q4: In Figure 4(b), it seems that in some cases the fairness-F1 scores can go below random. For example, in Figure 4(b) left, gender-F1 (binary classification) can be <0.5. This seems counter-intuitive, as the best fairness should be achieved at gender-F1=0.5.

I thus have a question, when gender-F1<0.5 (e.g. 0.45), can the attacker simply flip the prediction and get gender-F1>0.5 (e.g. 0.55)? What does it imply on fairness?

Q5: The authors state that "Intuitively, highly relevant data samples are likely to share the same fairness-sensitive feature" in line 178. I wonder after the model is trained well, and $\mathbf{s}$ does not contain fairness-related information, does this assumption still hold? If not, how will it influence the model training?

**Ethics Review Area:**

["I don’t know"]

**Limitations:**

I do not see any ethical concerns.

**Strengths And Weaknesses:**

**I am not very familiar with the fairness issue in machine learning. Please correct if I make misunderstandings regarding fair ML.**

Strengths:
1. **Well-motivated problem.** Trustworthy machine learning is a good topic to explore. The authors combine trustworthy machine learning with VFL, which is a good attempt.
2. Evaluations that verify the privacy protection (Fig. 3b) is a plus. It evaluates how well the proposed method protects attribute privacy, which is a focus in FL.
3. **Presentation is OK**. Despite the very complex gradient flow (in my opinion), the authors manage to explain the overall workflow to me.

Weaknesses:
1. **Lacking a baseline**. The authors take plain VFL as the only baseline in the feature-partition scenario. Although there are other fair ML methods compared (e.g. FairGO, FairSM), they cannot be applied in the VFL setting (as far as I can infer from the paper). Therefore, an intuitive question to ask is that, are there simpler VFL+fairness baselines that can be applied to the problem, such as slightly modifying FairSM, FairGO? If there is such a baseline, the improvements made by the authors' designs would be clearer.

2. **Definition of fairness**. The authors state in Sec. 3.1, line 114 that, "the counterfactual fairness requires that the model prediction for a sample should always hold if we only change fairness-sensitive attributes". However, in the evaluation, the authors use the metric "Gender F1", namely how well an attacker can guess the sensitive attribute, to measure fairness. I am a little confused about the metric. Specifically, as I can tell from Section 3.2, the unified representation $\mathbf{s}$ is computed from $\mathbf{s}_i^l$, which are all computed from fairness-insensitive features. In other words, sensitive attributes are not involved in the forward computation for the task $t$. It seems that in this sense, we have no chance to verify the 'counterfactual fairness', as we cannot even change the fairness-sensitive attributes. Thus, how the metrics (gender F1) are related to the notion of 'counterfactual fairness' should be better elaborated. Maybe the Gender F1 is more related to notions like conditional independence as far as I know.

3. **Should discuss limitations and tradeoffs on efficiency and stability**. The overall workflow of FairVFL is rather complex (even though the authors manage to explain it to me). As far as I am concerned, there are many points in the workflow of FairVFL that contain computation dependencies and may lead to stragglers (e.g. the iterative adversarial learning of $A$, $D$), which may significantly compromise the efficiency. Thus, I would like to see an efficiency comparison between naive VFL and FairVFL to see the tradeoffs between fairness and efficiency. Also, adversarial training tend to be very unstable. The authors should discuss how to mitigate the instability and generate consistent results.

---

> ### Author Response · Authors · 2022-08-02
> **Response to the Reviewer rSwq**
>
> Thank the reviewer for the insightful comments and constructive suggestions. Our detailed responses to your comments are as follows.
>
> **Q1:** Lacking a baseline.
>
> **A1:** The simple combination of existing fair ML methods [1,2] and the VFL framework can improve the model fairness of VFL tasks. However, these methods usually need to compute adversarial gradients from local sensitive attributes and may arouse privacy concerns, since data representations encoding private information need to be shared across sensitive attribute platforms. To improve the fairness of VFL models and meanwhile not hurt privacy, we apply adversarial learning in VFL for fair model training, and propose a contrastive adversarial learning method to protect user privacy (Section 3.3). Thus, the ablation of our FairVFL method (i.e., removing the contrastive adversarial learning method) can be viewed as the baseline method mentioned by the reviewer. The simple baseline can achieve comparable fairness with FairVFL and other centralized fair ML methods, however, it may lead to privacy leakage of privacy-sensitive attributes (Fig. 3(b)). We will discuss this issue in our revised paper.
>
> **Q2:** Definition of fairness.
>
> **A2:** Since the adversarial learning can help achieve counterfactual fairness in centralized machine learning [1], we apply it to improve the fairness of VFL models. However, as pointed out by the reviewer, it is difficult to directly verify the counterfactual fairness of different methods. Thus, following previous works [1,2], we evaluate the independence of fairness-sensitive features and data representations to verify the fairness of different methods. We will add these discussions to our revised paper.
>
> **Q3:** Should discuss limitations and tradeoffs on efficiency and stability.
>
> **A3:** Compared with the standard VFL framework, FairVFL only increases the computation cost of the server and platforms with fairness-sensitive attributes. The server and platforms need to extra compute the gradients on the attribute discriminators and contrastive discriminators, respectively. The computation complexities of FairVFL and standard VFL are $\mathcal{O}(d^3+D^3)$ and $\mathcal{O}(D^3)$, where $d$ and $D$ denote the parameter sizes of the discriminators and the main task model. In the widely used experimental settings, the discriminator is usually implemented by a small FFNN network, whose parameters are usually much less than the main task model. Thus, the computation efficiency of FairVFL is comparable with standard VFL methods. Besides, we agree that adversarial learning is sometimes unstable, which may affect fair model training. Fortunately, this problem has been studied by many previous works such as WGAN [3]. We can incorporate these works to improve the stability of FairVFL.
>
> **Q4:** In Figure 4(b), it seems that in some cases the fairness-F1 scores can go below random. When gender-F1<0.5 (e.g. 0.45), can the attacker simply flip the prediction and get gender-F1>0.5 (e.g. 0.55)? What does it imply on fairness?
>
> **A4:** A below-random fairness F1 score indicates that the model encodes the opposite biased information, which is also harmful to model fairness. This is because we evaluate the influence of the hyper-parameter $\gamma$ under a fixed hyper-parameter $\lambda$, where $\lambda$ and $\gamma$ control the importance of adversarial learning and contrastive adversarial learning methods respectively. When $\gamma$ is set to 0, the weight of the adversarial loss may be too large, and the adversarial gradients may make the model encode some opposite biased information. When we set the $\gamma$ to 0.25, the relative weight of adversarial learning is reduced, and the model fairness F1 score becomes close to the random guess.
>
> **Q5:** The authors state that "Intuitively, highly relevant data samples are likely to share the same fairness-sensitive feature" in line 178. I wonder after the model is trained well, and **s** does not contain fairness-related information, does this assumption still hold? If not, how will it influence the model training?
>
> **A5:** The assumption may not hold when the biased information is reduced from the unified data representations **$s$**. Fortunately, this issue will not damage the fair model training in FairVFL. This is because this assumption is used in contrastive adversarial learning to generate a protected representation that only encodes the biased information in **$s$** and reduces other private user information. The protected representation is further used to learn adversarial gradients to improve model fairness. When the assumption does not hold, the biased information is also reduced from the protected representation, which can make the adversarial learning converge.
>
> [1] Towards Personalized Fairness based on Causal Notion. SIGIR 2021.
>
> [2] Learning Fair Representations for Recommendation: A Graph-based Perspective. WWW 2021.
>
> [3] Wasserstein Generative Adversarial Networks. ICML 2017.

---

> > ### Comment · Reviewer_rSwq · 2022-08-07
> > **Author response acknowledged**
> >
> > I appreciate the authors' response on my questions. They more or less answer my questions. However, as I am not an expert in fairness-aware ML, I maintain my score at 6 at the present stage.

---

### Official Review · Reviewer_Tmvi · 2022-07-07

**Rating:** 6
**Confidence:** 3
**Soundness:** 3 good
**Presentation:** 2 fair
**Contribution:** 3 good

**Summary:**

This paper proposes a novel vertical federated learning framework (FairVFL) to achieve fair VFL models while keeping information private during communication. In order to encode a fair unified representation for the task platform, the prosed method employs adversarial learning at the fairness-insensitive platform side to prevent the fairness-sensitive bias. Moreover, instead of directly sending the unified representation to the fairness-insensitive platforms, the server sends a protected representation encoded by a mapper trained by a novel contrastive learning method.

**Questions:**

- How consistent are $\lambda$ and $\gamma$ cross datasets?

**Limitations:**

Yes, the authors mention the limitations in the appendix. As the authors mention, the method requires more rounds of communications than the baseline VFL. Meanwhile, it also requires more computations than the baseline.

**Strengths And Weaknesses:**

Strengths:
- The paper is well-written and organized. The method is clearly explained and well-motivated.
- The results are very convincing. The proposed method can have a slightly higher performance on fairness with a reasonable amount of drop in main task performance, compared to the methods under centralized training. Also, plenty of experiments on different models show the method is model-agnostic.
- The ablation study on adversarial learning and contrastive adversarial learning helps shed light on the power of the two methods in the framework.

Weaknesses:
- Multiple times of communications might be a problem in federated learning.
- Also, the local adversarial training might also slow down the overall training and increase the computational burden for both servers and platforms.
- Two hyperparameters, $\lambda$ and $\gamma$, seem to be very important. It might be hard for the server to choose the correct values for a new task.

---

> ### Author Response · Authors · 2022-08-02
> **Response to the Reviewer Tmvi**
>
> Thank the reviewer for the insightful comments and constructive suggestions. Our detailed responses to your comments are as follows.
>
> **Q1:** Multiple times of communications might be a problem in federated learning.
>
> **A1:** In a single training round, FairVFL requires two extra communication rounds between the fairness-sensitive platforms and the server, which may increase communication costs. Fortunately, the extra communication costs are usually small. This is because in the extra communication rounds FairVFL only needs to exchange protected representations and their gradients, and the communication cost is $\mathcal{O}(4BME)$, where $M$ is the number of fairness-sensitive platforms, $E$ denotes the dimensionality of protected representations, and $B$ denotes the batch size. Since $M$, $E$ and $B$ can be small in many settings, the corresponding cost is usually acceptable for data platforms. For example, in our experiments, $M$, $E$, and $B$ are set to 2, 64, and 32 respectively, and the extra communication cost of training model on a single sample is only 64kB.
>
> **Q2:** The local adversarial training might also slow down the overall training and increase the computational burden for both servers and platforms.
>
> **A2:** Compared with the standard VFL framework, FairVFL only increases the computation cost of the server and platforms with fairness-sensitive attributes. The server and platforms need to extra compute the gradients on the attribute discriminators and contrastive discriminators, respectively. The computation complexities of FairVFL and standard VFL are $\mathcal{O}(d^3+D^3)$ and $\mathcal{O}(D^3)$, where $d$ and $D$ denote the parameter sizes of the discriminators and the main task model. In the widely used experimental settings, the discriminator is usually implemented by a small FFNN network, whose parameters are usually much less than the main task model. Thus, the computation efficiency of FairVFL is comparable with standard VFL methods.
>
> **Q3:** Two hyperparameters, $\gamma$ and $\lambda$, seem to be very important. It might be hard for the server to choose the correct values for a new task. How consistent are $\gamma$ and $\lambda$ across datasets?
>
> **A3:** The optimal setting of $\gamma$ (around 0.25) is consistent across the two experimental datasets. Besides, the optimal setting of $\lambda$ for the two datasets is different (10 for ADULT dataset and 0.25 for NEWS dataset). Thus, the server may need to take some effort to search for the optimal setting of $\lambda$ on different datasets. In our future work, we plan to explore replacing the hyper-parameters with adaptive loss weights to avoid hyper-parameter tuning. We will discuss this limitation in the revised paper.

---

### Official Review · Reviewer_ajz9 · 2022-07-12

**Rating:** 6
**Confidence:** 3
**Soundness:** 3 good
**Presentation:** 3 good
**Contribution:** 3 good

**Summary:**

This paper proposed an approach to improve the fairness of VFL models without the requirement of storing the fairness-sensitive features in a central server. To improve the fairness of VFL models, an additional adversarial loss is designed to remove the bias of unified feature representation. The experiments result show the effectiveness of the proposed fairness algorithms for VFL.

**Questions:**

1). The authors claim that their approach can protect the exchange of information between the local model, aggregation models and task-specific models. The protection is based on removing the useful information from embedding. However, much research shows that the exchange of information may disclose the information on the original data, e.g. [1,2]. I am wonderring if the proposed training apporach can defende this attacks.

**Limitations:**

The variety of the dataset could be improved.

**Strengths And Weaknesses:**

Strengths:
1.) The paper is well-written and easy to understand the concepts and algorithms.
2). The research question is in-time and the fariness related research in VFL setting is less explored.
3). The experiment's result seems promising.

Weaknesses:
1). It would be better if the experiments can be conducted in a different types of datasets, e.g. image dataset celebA.

[1] Batch Label Inference and Replacement Attacks in Black-Boxed Vertical Federated Learning
[2] Deep Leakage from Gradients

---

> ### Author Response · Authors · 2022-08-02
> **Response to the Reviewer ajz9**
>
> Thank the reviewer for the insightful comments and constructive suggestions. Our detailed responses to your comments are as follows.
>
> **Q1:** It would be better if the experiments can be conducted in a different types of datasets, e.g. image dataset celebA.
>
> **A1:** In our work, we used two datasets with different modalities (i.e., tabular data and textual data) for evaluation. We agree with the reviewer that adding experiments on a new image dataset (CelebA) can better verify the effectiveness and generality of our work. Thus, we have followed the reviewer’s suggestion and compared different methods on CelebA. The detailed experimental settings are summarized as follows. To simulate the VFL setting, we use the raw images and a part of the raw attributes as the input features. The raw images are stored in a data platform and other input attributes are stored in another platform. The target task is to classify whether the input sample has the attribute “smiling”. We also treat gender (the “male’’ attribute) as the target fairness-sensitive attribute. The evaluation methods of model accuracy and fairness are the same as the experiments in the paper. Moreover, the model for the target task is the ResNet-10. We repeat each experiment for ten times and report the average results in the following table. The results show that FairVFL can still achieve comparable fairness-performance trade-off as centralized fair machine learning methods. We will add these results to the revised paper.
>
> |  Method  |   Task-Accuracy   |      Task-F1      |    Fairness-F1    |
> |--------|-----------------|-----------------|-----------------|
> | CenTrain | 0.9180$\pm$0.0013 | 0.9180$\pm$0.0014 | 0.8706$\pm$0.0069 |
> |  FairGo  | 0.9096$\pm$0.0086 | 0.9095$\pm$0.0088 | 0.5159$\pm$0.0570 |
> |  FairSM  | 0.9032$\pm$0.0230 | 0.9027$\pm$0.0242 | 0.5259$\pm$0.0807 |
> |  FairRec | 0.9070$\pm$0.0094 | 0.9067$\pm$0.0097 | 0.5241$\pm$0.0743 |
> |    VFL   | 0.9192$\pm$0.0020 | 0.9174$\pm$0.0021 | 0.8727$\pm$0.0056 |
> |  FairVFL | 0.9045$\pm$0.0130 | 0.9042$\pm$0.0135 | 0.5143$\pm$0.0778 |
>
>
> **Q2:** However, much research shows that the exchange of information may disclose the information on the original data, e.g. [1,2]. I am wondering if the proposed training approach can defend this attack.
>
> **A2:** We agree with the reviewer that the exchange of information (i.e., representations and gradients) across different platforms may arouse privacy concerns. In our method, we propose a contrastive adversarial learning method to protect private information in data representations, which can compare data representations with the same attribute and remove the attribute-irrelated information in them (More details are in Section 3.3). Experimental results in Fig.3(b) also verify that our method can effectively protect privacy without degrading model fairness and accuracy. To further protect private information in gradients, we can apply differential privacy techniques to perturb exchanged gradients as standard VFL methods [1].
>
> [1] Privacy Preserving Vertical Federated Learning for Tree-based Models. VLDB 2020.

---

### Meta-Review · Area_Chair_AFjT · 2022-08-26

**Recommendation:** Accept
**Confidence:** Less certain

**Metareview:**

This paper presents a fair vertical federated learning framework (FairVFL), by learning a set of unified fair representations of data/features distributed across decentralized platforms, i.e., these representations do not reflect sensitive attributes such as age/gender.  This is accomplished by having platforms learn local data representations from fairness-insensitive features, which are then aggregated at a central server; this is then sent (after a contrastive adversarial learning method removes private information) to platforms with fairness-sensitive features to remove bias by using adversarial learning.

All reviewers found the method sound and the experimental results convincing -- two real-world datasets were used to show that the proposed method can effectively improve model fairness while preserving user privacy.

Some concerns were raised over the communication and computational overhead of the proposed method, but none of the reviewers considered it serious enough as to merit rejection.

One note of caution on the Accept recommendation: this is a paper that did not receive a review with confidence level above 3.  Usually there is at least one review at the level of 4.

**Award:**

No

---

### Decision · Program_Chairs · 2022-09-14

Accept